# MULTI-SCALE FUSION FOR OBJECT REPRESENTATION

**Rongzhen Zhao[1], Vivienne Wang[1], Juho Kannala[2,3], Joni Pajarinen[1]**

[1]Department of Electrical Engineering and Automation, Aalto University, Finland

[2]Department of Computer Science, Aalto University, Finland

[3]Center for Machine Vision and Signal Analysis, University of Oulu, Finland

{rongzhen.zhao, vivienne.wang, juho.kannala, joni.pajarinen}@aalto.fi

## ABSTRACT

Representing images or videos as object-level feature vectors, rather than pixel-level feature maps, facilitates advanced visual tasks. Object-Centric Learning (OCL) primarily achieves this by reconstructing the input under the guidance of Variational Autoencoder (VAE) intermediate representation to drive so-called *slots* to aggregate as much object information as possible. However, existing VAE guidance does not explicitly address that objects can vary in pixel sizes while models typically excel at specific pattern scales. We propose *Multi-Scale Fusion* (MSF) to enhance VAE guidance for OCL training. To ensure objects of all sizes fall within VAE's comfort zone, we adopt the *image pyramid*, which produces intermediate representations at multiple scales; To foster scale-invariance/variance in object super-pixels, we devise *inter/intra-scale fusion*, which augments low-quality object super-pixels of one scale with corresponding high-quality super-pixels from another scale. On standard OCL benchmarks, our technique improves mainstream methods, including state-of-the-art diffusion-based ones. The source code is available on https://github.com/Genera1Z/MultiScaleFusion.

## 1 INTRODUCTION

According to vision cognition research, humans perceive a visual scene as objects and relations among them for higher-level cognition such as understanding, reasoning, prediction, planning and decision-making (Bar, 2004; Cavanagh, 2011; Palmeri & Gauthier, 2004). In computer vision, Object-Centric Learning (OCL) is a promising way to achieving similar effects – Images or video frames are represented as sparse object-level feature vectors, known as *slots*, rather than dense (super-)pixel-level feature maps under weak or self-supervision (Greff et al., 2019; Burgess et al., 2019). Meanwhile, segmentation masks of objects and the background are generated as byproducts, intuitively reflecting the corresponding object representation quality of these slots.

Object-level representation learning is highly influenced by textures. Early OCL methods, such as the mixture-based (Locatello et al., 2020; Kipf et al., 2022; Elsayed et al., 2022), which directly reconstruct the input pixels as supervision, often struggle with objects that have complex textures. To address this issue, mainstream OCL methods, such as transformer-based (Singh et al., 2022a;c), foundation-based (Seitzer et al., 2023; Zadaianchuk et al., 2024) and diffusion-based (Jiang et al., 2023; Wu et al., 2023b), leverage well-learnt intermediate representation for guided reconstruction, so as to drive slots to aggregate as much object-level information as possible.

Objects appear at different scales in images or videos due to imaging distances and actual sizes. Encoders/decoders usually perform well on certain pattern scales, necessitating "divide-and-rule" (Zou et al., 2023; Minaee et al., 2021). Mainstream OCL, however, overlooks such an issue.

Suppose three objects of different sizes in the input image or video frame. As shown in Fig. 1 upper left, current single-scale VAE encoding/decoding are likely only good at some specific scale of patterns, and only represent object 1 in high quality or confidence, leaving the other two being represented in low quality. Thus such single-scale VAE representation could not provide good guidance to OCL training on objects 2 and 3. But, as shown in Fig. 1 lower left and right, if resize the input into three scales, we ensure all the objects fall within the comfort-zone of the VAE encoder/decoder.

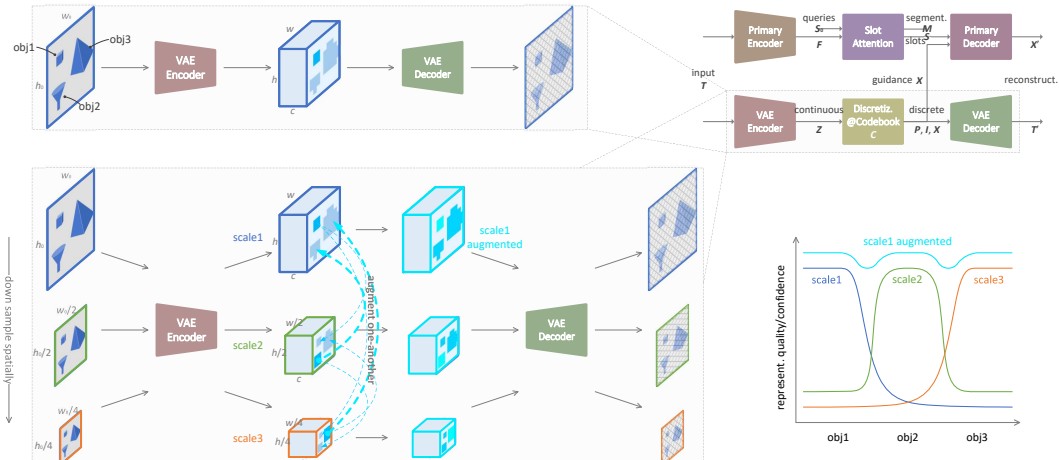

Figure 1: Multi-scale fusion is necessary in VAE guidance to OCL training. *Upper right*: simplified architecture of OCL with VAE guidance. *Upper left*: single-scale VAE encoding/decoding only represent patterns of some certain scale in high quality/confidence. *Lower left and right*: by resizing the input, patterns of different scales all fall within the VAE encoder/decoder's comfort-zone; low-quality super-pixels in one scale can be augmented by the high-quality in other scales.

Moreover, as shown in Fig. 1 lower left center, for scale 1, the low-quality super-pixels of object 2 and 3 can be augmented by their corresponding representations in scale 2 and 3, respectively.

We propose *Multi-Scale Fusion* (MSF) upon VAE representation, to guide OCL of both the transformer-based classics and diffusion-based state-of-the-arts. (*i*) We utilize technique image pyramid by resizing the input into different scales, to ensure objects of various sizes all fall within VAE encoder/decoder's *comfort zones*. (*ii*) We devise inter-/intra-scale quantized fusion by quantizing VAE representation of different scales with shared/specified codebooks, to foster *scale-invariance/-variance* in object representation. (*iii*) Our technique augments VAE discrete representation with better *object separability* and, in turn, provides better guidance for OCL training.

## 2 RELATED WORK

*VAE & OCL.* Pretrained VAE (Im Im et al., 2017; Van Den Oord et al., 2017), like dVAE or VQ-VAE, can provide representations that highlight object separability in the input by suppressing texture redundancies for OCL. With such guidance, transformer-based OCL methods (Singh et al., 2022a;c), like SLATE and STEVE, decode slots, which are extracted by SlotAttention (Locatello et al., 2020) from input images or video frames, into the input via a transformer decoder. Similarly, diffusion-based methods (Jiang et al., 2023; Wu et al., 2023b), like SlotDiffusion, decode slots via a diffusion model. Slots are driven to aggregate as much object information for good object representation. We improve these two lines of work. Foundation-based methods (Seitzer et al., 2023; Zadaianchuk et al., 2024), like DINOSAUR, are also included for complete comparison.

*Channel Grouping*. Widely adopted in CNNs (Zhang et al., 2018) and ViTs (Gu et al., 2022), grouping features along the channel dimension helps learn expressive representations. Grouping can be performed on features (Krizhevsky et al., 2012; Huang et al., 2018; Chen et al., 2019), or on weights (Zhao et al., 2021; 2022). SysBinder (Singh et al., 2022b) explores this in OCL by grouping slots queries to aggregate different object information, for emerging attributes under extra supervision. The most recent GDR (Zhao et al., 2024) groups the VAE codebook from features into combinatorial attributes, achieving better generalization and convergence in object representing. Our work also groups it but for scale-invariance and -variance.

*Multi-Scale*. Objects exist at different scales due actual sizes and imaging distances. CNN (Zhang et al., 2018) and ViT (Gu et al., 2022) typically have their comfort zones as the scale distribution of objects in training data is always nonuniform. In computer vision, like detection (Zou et al., 2023) and segmentation (Minaee et al., 2021), the multi-scale issue is significant, leading to the

wide adoption of "divide-and-rule". The image pyramid (Adelson et al., 1984) produces multi-scale representations by resizing; Feature pyramid network (Lin et al., 2017) and variants mix features of different layers by channel-concat and element-wise-sum. We are the first to realize multi-scale fusion on VAE, augmenting every scale with the high-quality representation in other scales.

## 3 PROPOSED METHOD

We propose Multi-Scale Fusion (MSF), to explicitly address the multi-scale issue in VAE intermediate representation, and thereby to guide Object-Centric Learning (OCL) better. Our technique is applicable to both transformer-based OCL (Singh et al., 2022a;c) classics and diffusion-based OCL (Wu et al., 2023b) state-of-the-arts, which are our *basis* methods. Note that two types of methods use dVAE (Im Im et al., 2017) and VQ-VAE (Van Den Oord et al., 2017) respectively, but we unify them with VQ-VAE like that in GDR (Zhao et al., 2024).

### 3.1 BACKGROUND: OCL WITH VAE GUIDANCE

Assume there is an input image or video frame $T \in \mathbb{R}^{h_0 \times w_0 \times c_0}$, where typically $h_0 = w_0 = 128$ and $c_0 = 3$.

Initially, the primary encoder encodes the input into a feature map. $\phi_e : T \to F$, where $F \in \mathbb{R}^{h \times w \times c}$, and $\phi_e$ is a CNN (He et al., 2016) or ViT (Caron et al., 2021) module. Typically, $h = w = 32$ and $c = 256$.

Then, the Slot Attention (SA) (Locatello et al., 2020) is used to aggregate the dense feature map $F$ into sparse object-level feature vectors, i.e., slots $S \in \mathbb{R}^{s \times c}$, along with the byproduct, i.e., segmentation masks $M \in \mathbb{R}^{s \times h \times w}$, under query vectors $S_0$. $\phi_{sa} : (S_0, F) \to (S, M)$, which is basically an iterative Query-Key-Value attention (Bahdanau et al., 2015) mechanism with $S_0$ as the query and $F$ as the key and value. Typically, $s$ is a predefined number, usually the maximum number of objects plus the background in an image or video sample of a dataset.

Meanwhile, pretrained VAE (Im Im et al., 2017; Van Den Oord et al., 2017) converts $T$ into continuous intermediate representation. $\phi_e^v : T \to Z$, where $Z \in \mathbb{R}^{h \times w \times c}$ and typically $h = w = 32$ and $c = 256$. This continuous representation is quantized into discrete representation by matching continuous representation with codebook and selecting matched codes to form discrete representation. $f_q = f_m \circ f_s : (Z, C) \to (P, I, X)$, where codebook $C \in \mathbb{R}^{m \times c}$, matching probabilities $P \in \mathbb{R}^{h \times w \times m}$, matched indexes $I \in \mathbb{R}^{h \times w}$ and discrete representation $X \in \mathbb{R}^{h \times w \times c}$. The matching $f_m$ and selection $f_s$ are implemented as below, respectively

$$P = \text{softmax}_m(-||Z - C||_2), \quad I = \text{argmax}_m(P) \tag{1}$$

$$X = \text{select}_m(C, I) \tag{2}$$

where each super-pixel $P^{(i,j,:)} \in \mathbb{R}^m$ means the matching probabilities of the $m$ template features in $C$. So the index of the maximum element in $P^{(i,j,:)}$, i.e., $I^{(i,j)}$ is the index of the most matched code, which can quantize $Z^{(i,j,:)}$ into $X^{(i,j,:)} := C^{(I^{(i,j)},:)}$.

Lastly, with $S$ as the condition, the primary decoder reconstructs the input guided by VAE discrete intermediate representation conditioned on $S$, which drives every slot to aggregate as much object information as possible. $\phi_d : (X, S) \to X'$, where $X' \in \mathbb{R}^{h \times w \times c}$. For transformer-based OCL, a transformer decoder (Vaswani et al., 2017) with internal masking performs the reconstruction as classification. For diffusion-based OCL, a diffusion model, typically a conditional UNet (Rombach et al., 2022), carries out the reconstruction as regression of the noise added to $X$.

For *VAE pretraining*, there is also VAE decoding from discrete representation to the input $\phi_d^v : X \to T'$, where $T' \in \mathbb{R}^{h_0 \times w_0 \times c_0}$. And the supervision signal comes from Mean-Squared Error (MSE) between $T$ and $T'$. For *OCL training*, the pretrained VAE module is frozen, and no VAE decoding occurs. Remaining parts are trained under the supervision signal of either Cross Entropy (CE) between $X$ and $X'$, or Mean-Squared Error (MSE) between the ground-truth noise added to $X$ and the reconstructed noise.

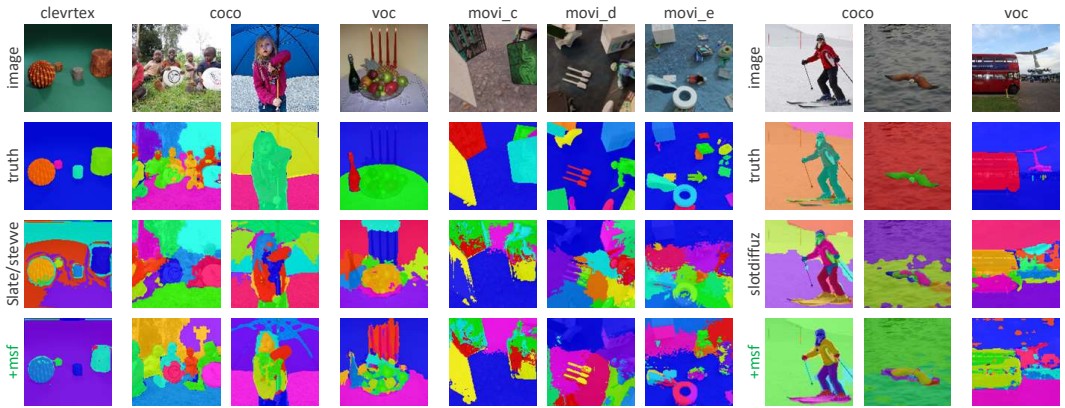

Figure 2: Qualitative results of OCL unsupervised segmentation. The last row is our MSF.

## 3.2 NAIVE MULTI-SCALE TRAINING

There can be objects of different sizes in images or videos, while models usually have best-at patterns scales due to non-uniform data distributions. This makes *image pyramid* (Adelson et al., 1984) an effective technique, where the input is resized into different sizes and processed in parallel. Such multi-scale representations make the pattern scale distribution more uniform for training, and ensures different sized objects to fall within models' comfort zone for testing.

Input image or video frame $T_1 := T$ is down-sampled with typical *bi-linear* interpolation into $T_n \in \mathbb{R}^{\frac{h_0}{2^{n-1}} \times \frac{w_0}{2^{n-1}} \times c_0}$, where $n = 1...N$ means different spatial scales and we use $N = 3$.

Multi-scale VAE encoding is $\phi_e^v : T_n \to Z_n$, where $Z_n \in \mathbb{R}^{\frac{h}{2^{n-1}} \times \frac{w}{2^{n-1}} \times c}$ and typically $h = w = 32$ and $c = 256$.

Multi-scale VAE quantization is $f_q = f_m \circ f_s : (Z_n, C) \to (P_n, I_n, X_n)$, where $P_n \in \mathbb{R}^{\frac{h}{2^{n-1}} \times \frac{w}{2^{n-1}} \times m}$, $I_n \in \mathbb{R}^{\frac{h}{2^{n-1}} \times \frac{w}{2^{n-1}}}$ and $X_n \in \mathbb{R}^{\frac{h}{2^{n-1}} \times \frac{w}{2^{n-1}} \times c}$. Here codebook $C$ is shared over all $N$ scales so as to facilitate scale-invariant pattern learning that differentiates among different object types. But this strong inductive bias can *lose* scale-variant information that differentiates among different object instances of the same type; Besides, no information exchange among the scales is a kind of information *waste* – The object pattern that is well represented in one scale should augment its poorly represented patterns in the other scales. We address this in the next subsection.

Multi-scale VAE decoding is $\phi_d^v : X_n \to T'_n$, where $T'_n \in \mathbb{R}^{\frac{h_0}{2^{n-1}} \times \frac{w_0}{2^{n-1}} \times c}$.

During VAE pretraining, there are $N$ sets of reconstruction and quantization losses. During OCL training, only scale $n = 1$ is used as guidance, thus the supervision is no different from the former subsection. Although multi-scale guidance can be used, it is always harmful in practice.

By following such recipe, hopefully at least one of the $N$ scales of representation provides high-quality representation to some sized objects.

Note: The original VQ-VAE has only $n = 1$ scale, which is a special case of our method.

## 3.3 INTER/INTRA-SCALE QUANTIZED FUSION

Based on the above multi-scale quantization, we further devise our novel fusion among the scales. *Inter-scale fusion* utilizes scale-invariance to augment poorly represented features in some scales with the well-represented features in other scales; *Intra-scale fusion* supplements with scale-variant details in the corresponding scale. *Scale-invariant* patterns differentiate among different object types, while *scale-variance* distinguishes among different object instances of the same type.

*Firstly*, we project VAE continuous representation of a certain scale $Z_n$ into two copies, $Z_n^{si}$ for scale-invariance and $Z_n^{sv}$ for scale-variance, respectively. These two will finally be combined back, thus we employ the *invertible project-up-project-down* design proposed in Zhao et al. (2024) to

maintain the inductive biases added in between.

$$\boldsymbol{Z}_n^{\mathrm{si}}, \boldsymbol{Z}_n^{\mathrm{sv}} = \mathrm{chunk}_{c,2}(\boldsymbol{Z}_n \cdot \mathrm{pinv}(W)) \tag{3}$$

where $\mathrm{pinv}(W)$ is pseudo-inverse of the project-down matrix in Eq. 5 for project-up here; $\mathrm{chunk}_{c,2}(\cdot)$ splits the tensor into two along the channel dimension.

*Secondly*, for the scale-variant continuous representation, we conduct quantization by a codebook *specific* to the current scale. $f_{\mathrm{q}} = f_{\mathrm{m}} \circ f_{\mathrm{s}} : (\boldsymbol{Z}_n^{\mathrm{sv}}, \boldsymbol{C}_n^{\mathrm{sv}}) \to (\boldsymbol{P}_n^{\mathrm{sv}}, \boldsymbol{I}_n^{\mathrm{sv}}, \boldsymbol{X}_n^{\mathrm{sv}})$. These non-sharing codebooks enforces scale-variant representation learning.

*Meanwhile*, for the scale-invariant continuous representation, we conduct *inter-scale fusion* over scale-invariant quantization of all scales. $f_{\mathrm{rf}} : \{(\boldsymbol{Z}_n^{\mathrm{si}}, \boldsymbol{C}^{\mathrm{si}})\} \to \{(\dot{\boldsymbol{P}}_n^{\mathrm{si}}, \dot{\boldsymbol{I}}_n^{\mathrm{si}}, \dot{\boldsymbol{X}}_n^{\mathrm{si}})\}$, where $n = 1...N$. The shared codebook enforces scale-invariant representation learning.

Now that all scales $\{\boldsymbol{X}_n^{\mathrm{si}}\}$ consist of codes in a shared codebook $\boldsymbol{C}^{\mathrm{si}}$, matching probabilities $\{\boldsymbol{P}_n^{\mathrm{si}}\}$ for the same (super-)pixel region under different scales indicate how confident the model is about the representation quality. We utilize this to augment the low-quality representation in some scales with the high-quality in other scales. Take scales $n-1$, $n$ and $n+1$ as an example:

1. Given any super-pixel $\boldsymbol{z}_n^{\mathrm{si}} := \boldsymbol{Z}_n^{\mathrm{si},(i,j,:)} \in \mathbb{R}^c$ in any scale $n$, find the corresponding region/super-pixel(s) in other scales, and mean them along all spatial dimensions as voting. For scale $n-1$, we choose $\boldsymbol{Z}_{n-1}^{\mathrm{si},(2i:2i+2,2j:2j+2,:)} \in \mathbb{R}^{2\times2\times c}$ and mean it into a vector $\boldsymbol{z}_{n-1}^{\mathrm{si}} \in \mathbb{R}^c$; For scale $n+1$, we choose $\boldsymbol{z}_{n+1}^{\mathrm{si}} := \boldsymbol{Z}_{n+1}^{\mathrm{si},(\lfloor\frac{i}{2}\rfloor,\lfloor\frac{j}{2}\rfloor,:)} \in \mathbb{R}^c$.

2. Match $\{\boldsymbol{z}_{n-1}^{\mathrm{si}}, \boldsymbol{z}_n^{\mathrm{si}}, \boldsymbol{z}_{n+1}^{\mathrm{si}}\}$ respectively with codebook $\boldsymbol{C}^{\mathrm{si}}$ as in Eq. 1, then we have corresponding matching probabilities $\{\boldsymbol{p}_{n-1}^{\mathrm{si}}, \boldsymbol{p}_n^{\mathrm{si}}, \boldsymbol{p}_{n+1}^{\mathrm{si}}\} \in \{\mathbb{R}^m\}$. These probabilities can be taken as the super-pixel quality of the same region in scales $\{n-1, n, n+1\}$.

3. Select the most matched code with the highest probability among all scales as the quantization $\dot{\boldsymbol{X}}_n^{\mathrm{si},(i,j,:)}$ of the super-pixel $\boldsymbol{Z}_n^{\mathrm{si},(i,j,:)}$. Specifically, for scales $\{n-1, n, n+1\}$, the most matched code index is

$$\dot{\boldsymbol{I}}_n^{\mathrm{si},(i,j)} = \mathrm{argmax}_m(\mathrm{max}_n(\{\boldsymbol{p}_{n-1}^{\mathrm{si}}, \boldsymbol{p}_n^{\mathrm{si}}, \boldsymbol{p}_{n+1}^{\mathrm{si}}\})) \tag{4}$$

with which we can determine $\dot{\boldsymbol{X}}_n^{\mathrm{si},(i,j,:)}$ as in Eq. 2.

*Lastly*, for each scale $n$, conduct the *intra-scale fusion* between the fused scale-invariant representation and scale-variant representation. $f_{\mathrm{af}} : (\dot{\boldsymbol{X}}_n^{\mathrm{si}}, \boldsymbol{X}_n^{\mathrm{sv}}) \to \boldsymbol{X}_n$. This combines the augmented scale-invariant and scale-variant copies back into the final discrete intermediate representation.

$$\boldsymbol{X}_n = \mathrm{concat}_c(\dot{\boldsymbol{X}}_n^{\mathrm{si}}, \boldsymbol{X}_n^{\mathrm{sv}}) \cdot W \tag{5}$$

where $\mathrm{concat}_c(\cdot)$ is channel concatenation; project-down matrix $W \in \mathbb{R}^{2c\times c}$ is trainable.

The above processes can be parallelized using PyTorch, as shown in Algo. 1 pseudo code.

### 3.4 RESOURCE CONSUMPTION ANALYSIS

*Codebook-related parameters.* The basis methods use a codebook of size $(m, c) = 4096 \times 256$, which amounts to $1.048$M. We use (1) invertiable projection to project the continuous representation into two copies (and project them back), up to $c \times 2c = 2^{17}$; (2) four groups of codebooks of size $(m^{0.5}, c) = (64, 256)$, one shared for $N = 3$ scale-invariant scales, three specifically for $N = 3$ scale-variant scales, up to $m^{0.5} \times c \times (1 + N) = 2^{16}$. Thus our codebook-related parameters are $0.196$M – we have $81.25\%$ fewer codebook parameters than the basis methods.

*Computation and memory consumption.* The exact calculation relates with the VAE encoder/decoder layers, like `Conv2d`, `GroupNorm`, `Mish`, etc., thus has too many details. But due to the $O(n^2)$ complexity nature of convolution layers, we can quickly get the estimation in ratio. Denote the computation of the original VAE as unit, then our $N = 3$ scales' computations would be $1$, $\frac{1}{4}$ and $\frac{1}{16}$, since the latter two has $2\times$ and $4\times$ down sampling rates. Thus in VAE pretraining, our technique requires roughly $31.25\%$ more computation. The memory consumption increase is similar.

|  | ARI | ARI$_{fg}$ | mIoU | mBO |
|---|---|---|---|---|
| ClevrTex | | | | |
| SLATE$_r$ | 20.56 | 68.14 | 34.11 | 34.57 |
| +SysBind@2 | 23.27 | 71.27 | 35.63 | 36.62 |
| +GDR@$g$2 | 34.46 | 73.38 | 37.42 | 36.69 |
| +MSF | 32.70 | 80.70 | 40.61 | 41.48 |
| COCO | | | | |
| SLATE$_r$ | 24.18 | 24.54 | 21.37 | 21.76 |
| +SysBind@2 | 25.71 | 24.97 | 21.46 | 22.01 |
| +GDR@$g$2 | 30.37 | 29.95 | 23.47 | 22.98 |
| +MSF | 30.95 | 30.47 | 23.33 | 23.85 |
| VOC | | | | |
| SLATE$_r$ | 11.64 | 15.65 | 15.64 | 14.99 |
| +SysBind@2 | 11.75 | 16.04 | 15.73 | 15.01 |
| +GDR@$g$2 | 13.20 | 17.49 | 16.46 | 16.65 |
| +MSF | 12.17 | 16.54 | 16.74 | 16.69 |

Table 1: Transformer-based *image* OCL performance on synthetic and real-world datasets.

|  | ARI | ARI$_{fg}$ | mIoU | mBO |
|---|---|---|---|---|
| MOVi-C | | | | |
| STEVE$_c$ | 52.20 | 31.16 | 16.74 | 19.05 |
| +SysBind@2 | 51.87 | 34.64 | 17.36 | 19.12 |
| +GDR@$g$2 | 60.14 | 35.79 | 20.01 | 21.95 |
| +MSF | 60.94 | 36.22 | 20.33 | 22.74 |
| MOVi-D | | | | |
| STEVE$_c$ | 35.71 | 50.24 | 19.10 | 20.88 |
| +SysBind@2 | 35.34 | 52.16 | 19.46 | 21.53 |
| +GDR@$g$2 | 40.37 | 52.47 | 20.42 | 22.64 |
| +MSF | 43.20 | 55.64 | 21.21 | 23.14 |
| MOVi-E | | | | |
| STEVE$_c$ | 28.00 | 52.06 | 18.78 | 20.48 |
| +SysBind@2 | 28.47 | 55.46 | 18.95 | 20.50 |
| +GDR@$g$2 | 34.17 | 53.21 | 19.47 | 20.76 |
| +MSF | 36.70 | 54.28 | 20.39 | 22.37 |

Table 2: Transformer-based *video* OCL on synthetic datasets.

|  | ARI | ARI$_{fg}$ | mIoU | mBO |
|---|---|---|---|---|
| ClevrTex | | | | |
| SlotDiffuz$_r$ | 64.21 | 26.50 | 31.51 | 32.44 |
| +GDR@$g$2 | 69.21 | 37.83 | 34.74 | 34.03 |
| +MSF | 71.47 | 38.08 | 35.06 | 35.67 |
| DINOSAUR | 60.74 | 45.75 | 30.48 | 32.56 |
| COCO | | | | |
| SlotDiffuz$_r$ | 36.54 | 35.67 | 22.08 | 22.75 |
| +GDR@$g$2 | 37.68 | 36.33 | 22.73 | 22.25 |
| +MSF | 37.63 | 36.99 | 22.32 | 22.87 |
| DINOSAUR | 33.24 | 33.35 | 22.01 | 21.93 |
| VOC | | | | |
| SlotDiffuz$_r$ | 16.97 | 14.33 | 15.71 | 16.02 |
| +GDR@$g$2 | 18.20 | 15.59 | 16.11 | 17.04 |
| +MSF | 19.40 | 15.69 | 16.37 | 16.76 |
| DINOSAUR | 16.00 | 18.48 | 15.94 | 16.37 |

Table 3: Diffusion-based *image* OCL on synthetic and real-world datasets. * Subscript "r" and "c" stand for random and condition query initialization, respectively.

|  | Inter-Cluster $\uparrow$ | | Intra-Cluster $\downarrow$ | |
|---|---|---|---|---|
|  | VQ-VAE | +MSF | VQ-VAE | +MSF |
| @c256 | | | | |
| ClevrTex | 0.534 | 0.786 | 0.043 | 0.038 |
| COCO | 0.746 | 1.095 | 0.184 | 0.093 |
| VOC | 0.633 | 0.978 | 0.191 | 0.101 |
| MOVi-C | 0.506 | 0.842 | 0.092 | 0.090 |
| MOVi-D | 0.511 | 0.725 | 0.089 | 0.088 |
| MOVi-E | 0.499 | 0.719 | 0.103 | 0.094 |
| @c4 | | | | |
| ClevrTex | 0.857 | 1.473 | 0.103 | 0.099 |
| COCO | 1.283 | 1.745 | 0.218 | 0.114 |
| VOC | 1.046 | 1.659 | 0.196 | 0.132 |

Table 4: Object separability of VAE guidance. "Inter-Cluster" is the mean distance of super-pixels within a cluster, while "Intra-Cluster" is that among different clusters. "@c256" and "@c4" are VAE intermediate dimensions, for SLATE/STEVE and SlotDiffusion, respectively.

## 4 EXPERIMENTS

We experiment the following points: (*i*) Our technique MSF augments performance of mainstream OCL methods that are either transformer-based or diffusion-based. (*ii*) MSF fuses multi-scale information into VAE discrete representation and guides OCL better. (*iii*) How the composing designs of MSF contribute to its effectiveness. Results are mostly averaged over three random seeds.

### 4.1 OCL PERFORMANCE

To evaluate the quality of OCL object representations, i.e., slots, we use the accuracy of OCL's byproduct (unsupervised) segmentation as an intuitive measurement. Metrics including ARI (Adjusted Rand Index)[1], ARI$_{fg}$ (foreground), mIoU (mean Intersection-over-Union)[2] and mBO (mean Best Overlap) (Caron et al., 2021) are used to measure OCL's byproduct segmentation accuracy.

---

[1] https://scikit-learn.org/stable/modules/generated/sklearn.metrics.adjusted_rand_score.html
[2] https://scikit-learn.org/stable/modules/generated/sklearn.metrics.jaccard_score.html

Datasets are either synthetic images, i.e., ClevrTex[3]; or real-world images, i.e., COCO[4] and VOC[5]; or synthetic videos, i.e., MOVi-C/D/E[6].

To evaluate how MSF boosts OCL, we use transformer-based classics, SLATE (Singh et al., 2022a) for images and STEVE (Singh et al., 2022c) for videos; use diffusion-based state-of-the-arts, Slot-Diffusion (Wu et al., 2023b) for images. The foundation-based DINOSAUR (Seitzer et al., 2023) and competitors SysBinder (Singh et al., 2022b) and GDR (Zhao et al., 2024) are compared too. All primary encoders are unified as DINO (Caron et al., 2021) to form strong baselines.

Our MSF is a general improver to both transformer-based classics and diffusion-based state-of-the-arts, as shown in Tab. 1, 2 and 3. On complex synthetic dataset ClevrTex, MSF improves SLATE significantly, up to 12% in $ARI_{fg}$. On challenging real-world dataset COCO, MSF still manages to improve SLATE by 6% in $ARI_{fg}$. MSF even boosts SlotDiffusion, which is state-of-the-art, by 3% in ARI on VOC. Compared with other improvers, i.e., SysBinder@$g2$ and GDR@$g2$, our MSF also beats or at least ties them on SLATE, while beats them consistently on STEVE. On SlotDiffusion, our MSF beats the strong competitor GDR@$g2$, and defeats DINOSUAR.

By the way, our MSF boosts OCL performance on COCO and VOC less than on other datasets. We attribute this to those object sizes are often out of the coverage of scales 1~3. Namely, many humans, trees and grass lands in those real-world images take up the whole scene, and either down sampling $4\times$ (scale 1) or $16\times$ (scale 3) cannot produce scale-invariant patterns for inter-scale fusion.

## 4.2 VAE GUIDANCE

To investigate how MSF fuses multi-scale information into VAE guidance, we analyze *object separability* (Stanic et al., 2024; Lowe et al., 2024). Empirically, better object separability in VAE guidance contributes to better OCL performance. We cluster on the VAE guidance $\boldsymbol{X}_{n=1}$ then calculate inter-/intra-cluster distances; We also visualize (Caron et al., 2021) object separability of scale-invariant representations $\{\dot{\boldsymbol{X}}_n^{\mathrm{si}}\}$ and scale-variant representations $\{\boldsymbol{X}_n^{\mathrm{sv}}\}$, as well as representations before inter-scale fusion $\{\boldsymbol{X}_n^{\mathrm{si}}\}$ and representations after intra-scale fusion $\{\boldsymbol{X}_n\}$.

As shown in Fig. 3 and Tab. 4, our MSF contributes to significant object separability boosts both qualitatively and quantitatively upon basis methods. In contrast, the object separability of basis methods is consistently inferior to ours.

## 4.3 ABLATION STUDY

We use SLATE on ClevrTex as an example to evaluate how different designs contribute to our MSF. We use ARI+$ARI_{fg}$ as the metrics because ARI is mostly dominated by background segmentation performance while $ARI_{fg}$ only measures the foreground. We employ naive CNN (Kipf et al., 2022) as the primary encoder to reduce experiment time.

*Inter/intra-scale fusion*: with vs without. As shown in Tab. 5 upper left, both types of fusion are crucial to performance. Yet using inter-scale fusion alone can be even worse than disabling both, meaning that inter-scale fusion enforces scale-invariance at the cost of losing too much information, wheras intra-scale fusion can recover it. Therefore, enabling both types of fusion is the best setting.

*Scale-invariant/variant codebook*: shared vs specified. Firstly, using different codebooks as the scale-invariant codebook is meaningless because the corresponding matching probabilities for the same super-pixel cannot be compared. As shown in Tab. 5 upper right, sharing one codebook over different scales for the scale-variance is harmful to the performance. Hence different sets of codes are necessary to foster diverse patterns that are different among the scales.

*Number of scales*: 2, 3 or 4. As shown in Tab. 5 lower left, the default number of scales $N = 3$ is the best setting. By contrast, $N = 2$ includes too less explicit scales; As for $N = 4$, its last scale, i.e., scale 4, down samples too much and loses too much spatial information to support the inter-scale fusion of our MSF technique.

---

[3]https://www.robots.ox.ac.uk/~vgg/data/clevrtex/
[4]https://cocodataset.org/#panoptic-2020
[5]http://host.robots.ox.ac.uk/pascal/VOC/voc2012/index.html
[6]https://github.com/google-research/kubric/tree/main/challenges/movi

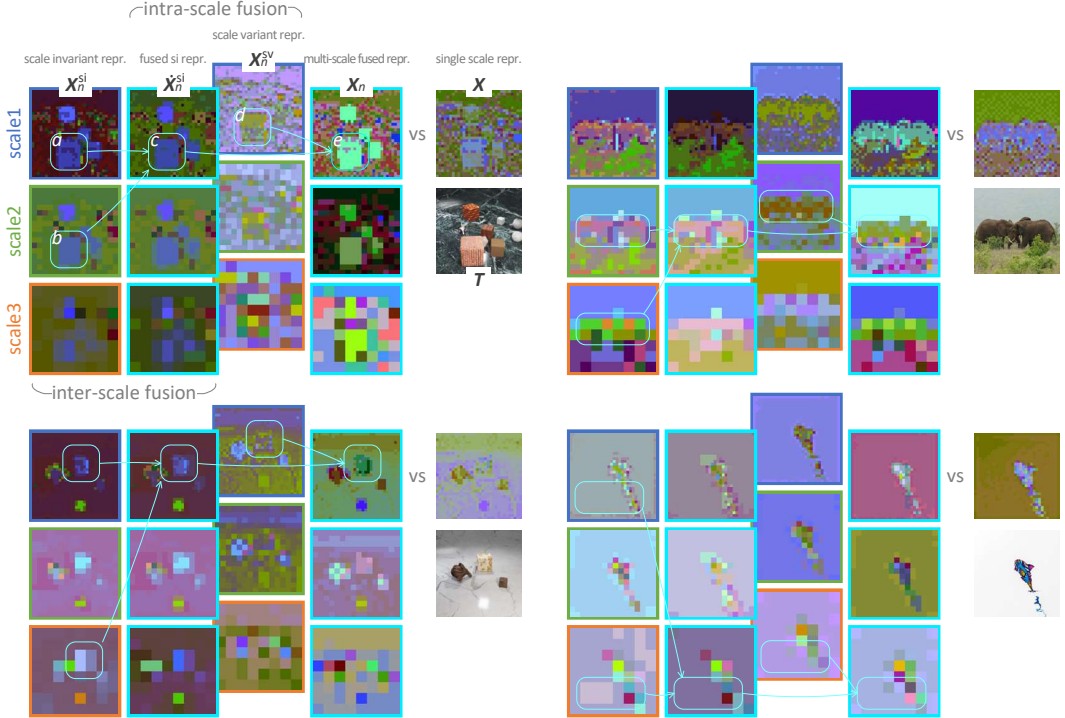

Figure 3: Qualitative object separability in VAE guidance with MSF. Take the upper left as an example. The circled cube $a$ in column $\boldsymbol{X}_n^{\text{si}}$ row scale1 is vague and noisy, while its corresponding representation $b$ in row scale2 is very clear, thus $b$ can augment $a$, yielding clearer $c$ in column $\dot{\boldsymbol{X}}_n^{\text{si}}$ row scale1. As $c$ only captures scale-invariant information, $d$ in column $\boldsymbol{X}_n^{\text{sv}}$ row scale1 supplements scale-variant information, which has too many details and is not object-like. By combining $c$ and $d$, we obtain $e$ in column $\boldsymbol{X}_n$ row scale1, making this cube much more separable from the background than in column $X$ the corresponding area. Notations are detailed in Sect. 4.2 beginning.

| inter-scale fuz | intra-scale fuz | ARI+ARI$_{\text{fg}}$ |
|:---:|:---:|:---:|
| ✓ | ✓ | 100.71 |
| ✓ | | 90.85 |
| | ✓ | 95.64 |
| | | 91.03 |

| scale-variant codebook(s) | | |
|:---:|:---:|:---:|
| specified | shared | ARI+ARI$_{\text{fg}}$ |
| ✓ | | 100.71 |
| | ✓ | 92.28 |

| number of scales | ARI+ARI$_{\text{fg}}$ |
|:---:|:---:|
| 2 | 95.83 |
| 3 | 100.71 |
| 4 | 98.96 |

| number of guidance | | |
|:---:|:---:|:---:|
| scale 1 | scale 2 | ARI+ARI$_{\text{fg}}$ |
| ✓ | | 100.71 |
| ✓ | ✓ | 96.42 |
| | ✓ | 89.59 |

Table 5: Ablation study of SLATE+MSF on ClevrTex using naive CNN as the primary encoder. (Upper left) effects of inter-scale fusion and intra-scale fusion; (Upper right) effects of specified or shared codebooks for scale-variance; (Lower left) effects of number of scales being used in VAE pretraining; (Lower right) effects of number of VAE guidance being used for OCL training.

*Number of guidance*: 1 or 2. After MSF in VAE, we have three augmented scales. But as shown in Tab. 5 lower right, using the augmented scale 1 as VAE guidance is the best choice, compared with either using both augmented scale 1 and 2, or using augmented scale 2 solely. This is because the augmented scale 1 representation has the largest resolution, i.e., the most spatial details, and also multiple scales of information. Although the other two scales are also multi-scale fused, the information incompleteness due to their low resolution is harmful to guide OCL training.

## 5 CONCLUSION

We propose a technique named MSF to improve existing OCL methods, including both transformer-based classics and diffusion-based state-of-the-arts. Our technique is realized by integrating the naive image pyramid with our novel inter-scale quantized fusion and intra-scale quantized fusion. Our technique ensures different sizes of objects are all processed in VAE models' comfort zone, and fosters scale-invariance and scale-variance in object representation. This is achieved at acceptable costs of extra computation and memory. We evaluate our technique via comprehensive experiments, and also interpret the multi-scale fusion processes by visualizing it. Such multi-scale fusion is also worth exploring in the primary encoder and slot attention part, or, in VAE designing for generative models, which are suggested for future work.

### ACKNOWLEDGMENTS

We acknowledge the support of the Finnish Center for Artificial Intelligence (FCAI) and the Research Council of Finland through its Flagship program. Additionally, we thank the Research Council of Finland for funding the projects ADEREHA (grant no. 353198), BERMUDA (362407) and PROFI7 (352788). We also appreciate CSC – IT Center for Science, Finland, for granting access to the LUMI supercomputer, owned by the EuroHPC Joint Undertaking and hosted by CSC (Finland) in collaboration with the LUMI consortium. Furthermore, we acknowledge the computational resources provided by the Aalto Science-IT project through the Triton cluster.

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

# A APPENDIX

## A.1 EXTENDED RELATED WORK

**Multi-scale & image/feature pyramid**. Existing multi-scale methods (i) Either employ an image pyramid (Adelson et al., 1984; Singh & Davis, 2018; Najibi et al., 2019) without information fusion

among pyramid levels, wasting useful information among different scales; (ii) Or rely on feature pyramids, namely, FPN and numerous variants (Lin et al., 2017; Tan et al., 2020; Piao et al., 2021), where features of different DNN depths are fused by channel-concat or element-wise-sum, mixing both low- and high-quality representations together. In contrast, *our MSF is the first to enable fusion of multiple scales on VAE representations*. By leveraging codebook matching (Van Den Oord et al., 2017), we selectively fuse high-quality information among scales, rather than mixing them together. As shown in Tab. 6, our MSF is superior to naive channel-concat or element-wise-sum.

|  | channel-concat | element-wise-sum | our msf |
|---|---|---|---|
| ARI+ARI$_{fg}$ | 92.63 | 89.18 | 100.71 |

Table 6: Effects of different fusions among scales. Model is SLATE; dataset is ClevrTex.

**Multi-scale & VAE**. Recent works like VAR (Tian et al., 2024) and SPAE (Yu et al., 2024) also build the multi-scale representations upon VAE. But they are different from our MSF as follows. (*i*) VAR vs MSF: VAR auto-regresses from smaller scales of VAE representation to the larger, but with no information fusion among scales. In contrast, our MSF realizes fusion among different scales on VAE for the first time. (*ii*) SPAE vs MSF: SPAE relies on multi-modal foundation model CLIP, while our MSF does not. SPAE element-wisely averages multiple scales into one, mixing both low- and high-quality representations together, i.e., no fusion among scales. Our MSF augments any scaled representation with all other scales' high-quality representations, i.e., fusion among scales.

**Relationships among SSLseg, OCL and WMs**. As shown in Fig. 4, (1) *SSLseg* (Self-Supervised Learning segmentation), e.g., HCL (Ge et al., 2023) and VideoCutLER (Wang et al., 2024), focuses on extracting segmentation masks; (2) *OCL* (Object-Centric Learning), e.g., SLATE (Singh et al., 2022a) and SlotDiffusion (Wu et al., 2023b), represents each object as a feature vector, with segmentation masks as byproducts, kind of overlapping with SSLseg; (3) *WMs* (World Models), e.g., SlotFormer (Wu et al., 2023a), upon OCL, address downstream tasks like visual reasoning, planning and decision-making.

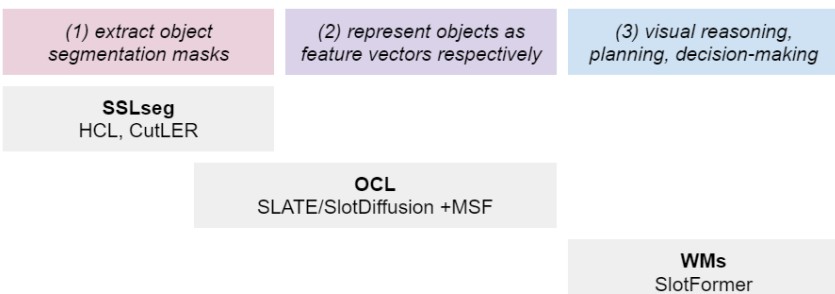

Figure 4: SSLseg vs OCL vs WMs.

## A.2 EXTENDED EXPERIMENTS

**MSF's effect on different sized objects**. We follow COCO's small/medium/large size splits to evaluate our MSF's performance. Results of resolution 128x128 are shown below Tab. 7. Our MSF does show better performance on small objects.

|  | mIoU$_S$ | mIoU$_M$ | mIoU$_L$ | overall |
|---|---|---|---|---|
| SLATE | 8.57 | 26.65 | 34.57 | 26.94 |
| +MSF | 12.63 | 28.14 | 34.83 | 29.57 |

Table 7: How MSF peforms on different sized objects. Dataset is COCO instance segmentation.

**MSF on OpenImages subset with higher resolution**. We use input resolutions 128 and 256 to evaluate on OpenImages subset Ge et al. (2023). Under resolution 128, we use $n$=3 and 4; under resolution 256, we use $n$=3, 4 and 5. Results are shown below Tab. 8. We use ARI+ARIfg as

the metrics because ARI mostly reflect how well the background is segmented and ARIfg only measures foreground objects. According to the results, under resolution 128, n=3 is the best choice; under resolution 256, n=4 is the best choice.

| resolution | 128×128 | | 256×256 | | |
|---|---|---|---|---|---|
| $n$ | 3 | 4 | 3 | 4 | 5 |
| ARI+ARI$_{\mathrm{fg}}$ | 62.07 | 60.93 | 64.82 | 67.58 | 62.75 |

Table 8: Effects of input resolution and #scales $n$. Model is SLATE+MSF; dataset is OpenImages subset.

## A.3 Model Architecture

The overall model structure is depicted in Fig. 1 upper right. It consists of a primary encoder, SlotAttention and primary decoder, as well as VAE encoder, codebook, and VAE decoder.

**Primary Encoder**

Unlike exiting works, which used a naive CNN or small ResNet, we use a pretrained foundation model DINO as the primary encoder on all datasets, so that we setup strong baseline to show our MSF's effectiveness.

To save computation while maintain strong performance for all methods, we adopt the big-little design used in (Zhao et al., 2024), which takes small resolution 128 as input while still produce high resolution (128) feature maps. We employ the DINO v1, tiny, with a downsampling rate of 8, as presented in (Caron et al., 2021). We integrate this into a big-little architecture by running DINO in parallel with the previously mentioned naive CNN. The output of DINO is then projected, upsampled, and combined element-wise with the output of the CNN. The architecture is briefly represented as dino()-conv(k3s1p1)-upsample(k8) + naivecnn(). Notably, the weights of DINO are frozen, which retains its pretrained features while allowing the CNN to focus on refining high spatial details.

Hence, no spatial downsampling occurs within this module. This part remains consistent across all experiments.

**SlotAttention**

We adhere to the standard design outlined in (Locatello et al., 2020). This design is consistent across all experiments utilizing this module.

**Primary Decoder**

This module can be either a transformer decoder or a diffusion model.

For transformer-based OCL, e.g., SLATE and STEVE, we follow the architecture described in (Singh et al., 2022a), unified across all experiments involving this component. This decoder operates on VAE intermediate feature tensors, which undergo 4x4 spatial downsampling relative to the input. SlotAttention-derived slots are provided as the conditioning input. The input channel dimension for this module is set to 256.

For diffusion-based OCL, e.g., SlotDiffusion, we follow the architecture described in (Wu et al., 2023b). This decoder take VAE representation with noise as input, taking slots, time step as condition, finally produce the reconstruction of the noise being added to the input. The input channel dimension for this module is set to 4.

**VAE Encoder & Decoder**

The architecture of this module is in accordance with the design described in (Singh et al., 2022a), and it is unified for all experiments incorporating this module. The VAE encoder applies a 4x4 spatial downsampling, and the VAE decoder performs a corresponding 4x4 upsampling.

**Codebook**

This module distinguishes between dVAE and VQ-VAE and is utilized in both transformer-based and diffusion-based methods.

---

**Algorithm 1** Implementation of inter/intra-scale quantized fusion in PyTorch.

```python
def ms_fuz(self_codebook: ModuleList, encodes: list, project: Module) -> list:
    s = len(encodes)
    assert len(self_codebook) == 1 + s  # shared*1 + specified*s
    b, c, h, w = encodes[0].shape

    encodes = [project(_, pinv=True) for _ in encodes]  # s*(b,c,h,w)->s*(b,2c,h,w)

    zidx1s, encode1s, quant1s = [], [], []
    for i1 in range(s):
        encode1_ = []
        for j1, encode in enumerate(encodes):
            if j1 == i1:
                pass
            elif j1 < i1:
                e1_ = avg_pool2d(encode[:, :c, :, :], 2 ** (i1 - j1))
            else:
                e1_ = upsample(encode[:, :c, :, :], scale_factor=2 ** (j1 - i1), mode="nearest")
            encode1_.append(e1_)  # s*(b,c,h,w)

        encode1_ = stack(encode1_)  # (s,b,c,h,w)
        zsoft1_, zidx1_ = self_codebook[0].match(encode1_.flatten(0, 1))
        zsoft1_ = zsoft1_.unflatten(0, [s, b])  # (s*b,m,h,w)->(s,b,m,h,w)
        # zidx1_ = zidx1_.unflatten(0, [s, b])  # (s*b,h,w)->(s,b,h,w)

        zsoft11, zidx11 = zsoft1_.max(2)  # (s,b,m,h,w)->(s,b,h,w)
        zidx12 = zsoft11.argmax(0)  # (s,b,h,w)->(b,h,w)
        zidx1 = zidx11.gather(0, zidx12[None, :, :, :])[0]
        # (s,b,c,h,w)->(b,c,h,w)
        encode1 = encode1_.gather(0, zidx12[None, :, None, :, :].expand(-1, -1, c, -1, -1))[0]
        quant1 = self_codebook[0](zidx1).permute(0, 3, 1, 2)

        zidx1s.append(zidx1)  # s*(b,h,w)
        encode1s.append(encode1)  # s*(b,c,h,w)
        quant1s.append(quant1)  # s*(b,c,h,w)

    zidx2s, encode2s, quant2s = [], [], []
    for i2 in range(s):
        encode2 = encodes[i2][:, c:, :, :]
        zsoft2, zidx2 = self_codebook[1 + i2].match(encode2)
        quant2 = self_codebook[1 + i2](zidx2).permute(0, 3, 1, 2)
        zidx2s.append(zidx2)  # s*(b,h,w)
        encode2s.append(encode2)  # s*(b,c,h,w)
        quant2s.append(quant2)  # s*(b,c,h,w)

    zidxs = [stack([u, v], 1) for u, v in zip(zidx1s, zidx2s)]  # s*(b,2,h,w)
    # encodes = [cat([u, v], 1) for u, v in zip(encode1s, encode2s)]  # s*(b,2c,h,w)
    quants = [cat([u, v], 1) for u, v in zip(quant1s, quant2s)]  # s*(b,2c,h,w)

    quants = [project(_) for _ in quants]  # s*(b,c,h,w)
    return zidxs, quants
```

---

For the non-grouped codebook design, we adhere to the typical dVAE and VQ-VAE implementations as described in (Singh et al., 2022a) and (Wu et al., 2023b), respectively. For the grouped codebook design used in SLATE/STEVE and SlotDiffusion, we take the GDR as the basis, by applying with our design as mentioned in main body of this paper. Thus except what we describe in the main body, we also adopt annealing residual connection and Gumbel sampling code matching to stabilize training.

Implementation of our MSF described in Sect. 3.3 is described in Code 1.

This module operates on VAE intermediate feature tensors with a 4x4 spatial downsampling relative to the input.

The non-grouped codebook contains 4096 codes; Similarly, the grouped codebook contains 4096 codes combined from scale-invariant and scale-variant codebooks, for which we introduce code replacing trick, instead of utilization loss, to improve code utilization. In transformer-based models, the code dimension (VAE intermediate channel size) is 256, whereas in diffusion-based models, the code dimension is 4.

**Basis Methods**

For SLATE/STEVE, we primarily adhere to the original designs for transformer-based models, SLATE and STEVE. Their primary encoder, SlotAttention, decoder, and VAE conform to the respective designs mentioned earlier. SLATE is tailored for image-based OCL tasks, while STEVE extends SLATE to video-based OCL tasks by adding an additional transformer encoder block to bridge current and subsequent queries.

For SlotDiffusion, we also follow its original design for diffusion-based methods. Its primary encoder, SlotAttention, decoder, and VAE conform to the designs outlined earlier. For video-based OCL tasks, we extend SlotDiffusion by adding an extra transformer encoder block to connect current and future queries. For the diffusion decoder, we use default noise scheduler and beta values.

**DINOSAUR**

DINOSAUR is included as a foundational method for OCL, serving primarily as a reference. We adopt its original design but unify its primary encoder and SlotAttention with those described above.

**Competitive Improvers**

SysBinder is one of the competitors. This module enhances the SlotAttention module in SLATE/STEVE. We follow its original design while keeping the other components consistent with the earlier-described modules. We use group number two.

GDR is another competitors. We use group sizes of two, and leave all the other settings identical to the original paper.

A.4   DATASET PROCESSING

The datasets used for image OCL include ClevrTex, COCO, and VOC, with the latter two being real-world datasets. For video OCL, the datasets are MOVi-C, D, and E, all of which are synthetic. While the processing is mostly consistent across datasets, there are a few differences.

**Shared Preprocessing**

To speed up experiments, we implement a dataset conversion approach: converting all datasets into the LMDB database format and storing them on an NVMe or RAM disk to reduce I/O overhead and maximize throughput.

In particular, all images or video frames, along with their segmentation masks, are center-cropped and resized to 128x128. These are then stored in the uint8 data type to save space. For images and videos, we apply the default interpolation mode, whereas for segmentation masks, we use the NEAREST-EXACT interpolation mode.

During training, for both images and videos, we use an input and output spatial resolution of 128x128. Input images and videos are normalized by subtracting 127.5 and dividing by 127.5, ensuring all pixel values fall within the range of -1 to 1. For videos, we apply random strided temporal cropping with a fixed window size of 6 to speed up training and improve generalization.

During testing, image processing remains identical to the training phase. However, for videos, we omit the random strided temporal cropping and instead use the full 24 time steps.

**ClevrTex, COCO & VOC**

For ClevrTex, each image can contain up to 10 objects, so we use 10 + 1 slot queries, with the extra slot representing the background.

Microsoft COCO and Pascal VOC are both real-world datasets. For COCO, we use panoptic segmentation annotations, as OCL favors panoptic segmentation over instance or semantic segmentation. However, for VOC, we rely on semantic segmentation annotations due to the absence of

panoptic or instance segmentation labels. Given that images can contain numerous objects, including very small ones, and considering the 128x128 resolution, we filter out images with more than 10 objects or objects smaller than a 16-pixel bounding box area. As a result, we use a maximum of 10 objects (plus stuff in COCO) per image, corresponding to 11 slot queries.

**MOVi-C/D/E**

These datasets provide rich annotations, such as depth, optical flow, and camera intrinsic/extrinsic parameters. However, for simplicity and comparison purposes, we limit our use to segmentation masks. Additionally, we retain the bounding box annotations for all objects across all time steps, as this conditioned query initialization is crucial and widely used in video OCL. Given that these datasets contain 10, 20, and 23 objects, respectively, we use 10+1, 20+1, and 20+1 slot queries.

## A.5 TRAINING SCHEME

In line with established practice, our training process comprises two stages. The first stage is pretraining, during which VAE modules are trained on their respective datasets to acquire robust discrete intermediate representations. In the second stage, OCL training utilizes the pretrained VAE representations to guide object-centric learning.

**Pretraining VAE**

Across all datasets, we conduct 30,000 training iterations, with validation every 600 iterations. This gives us roughly 50 checkpoints for each OCL model on every dataset. To optimize storage, we retain only the final 25 checkpoints. This approach is uniformly applied across datasets.

For image datasets, the batch size is set to 64, while for video datasets, it is 16, a configuration maintained across both training and validation phases. This is consistent for all datasets. We employ 4 workers for multi-processing, applicable to both training and validation. This configuration is the same for all datasets.

We use the Adam optimizer with an initial learning rate of 2e-3, adjusting the learning rate through cosine annealing scheduling, with a linear warmup over the first 1/20 of the total steps. This configuration is standardized across all datasets.

Automatic mixed precision is utilized, leveraging the PyTorch autocast API. In tandem with this, we use PyTorch's built-in gradient scaler to enable gradient clipping with a maximum norm of 1.0. This setting is uniform across all datasets.

**Initializing Slot Queries**

The query initializer provides the initial values to aggregate the dense feature map of the input into slots representing different objects and the background.

For image datasets like ClevrTex, COCO, and VOC, we use random query initialization. This method involves learning a set of Gaussian distributions and sampling from them. There are two parameters involved: the means of the Gaussian distributions, which are trainable and of dimension $c$, and the shared sigma (a scalar value), which is non-trainable and follows a cosine annealing schedule, fixed at 0 during evaluation. Both parameters are of the same dimension as the slot queries.

For video datasets such as MOVi-C/D/E, initialization takes a different approach. Prior knowledge, such as bounding boxes, is projected into vectors. The bounding boxes are normalized by the input frame dimensions and flattened into a 4-dimensional vector corresponding to the number of slots. This vector is then processed through a two-layer MLP, using GELU activation, to map the bounding boxes into the channel dimension $c$, consistent with the slot queries.

**Training the OCL Model**

On this stage, we load the pretrained VAE weights to guide the OCL model, where the VAE part is frozen.

For all datasets, we run 50,000 training iterations, validating every 1,000 iterations. This results in about 50 checkpoints per OCL model for each dataset. To reduce storage demands, only the last 25 checkpoints are saved. This procedure is applied across all datasets.

The batch size for image datasets is set to 32 for both training and validation. For video datasets, the batch size is 8 for training and 4 for validation, to account for the increased time steps during video validation. This setting is shared across datasets.

We utilize 4 workers for multi-process operations, consistent across both training and validation phases, and applied uniformly across datasets.

The metrics used for evaluation include ARI, mIoU, and mBO, which calculate the panoptic segmentation accuracy for both objects and background. These metrics are applied consistently across all datasets.

We use the Adam optimizer with an initial learning rate of 2e-4, following a cosine annealing schedule and a linear warmup over the first 1/20 of the total steps. This configuration is standardized across datasets.

We employ automatic mixed precision using the PyTorch autocast API. Alongside this, we use the PyTorch gradient scaler to apply gradient clipping, with a maximum norm of 1.0 for images and 0.02 for videos.

For random query initialization, we adjust the $\sigma$ value of the learned non-shared Gaussian distribution to balance exploration and exploitation. On multi-object datasets, $\sigma$ starts at 1 and decays to 0 by the end of training using cosine annealing scheduling. On single-object datasets, $\sigma$ remains at 0 throughout training. During validation and testing, this value is set to 0 to ensure deterministic performance.

