# OpenReview forum: "Multi-Scale Fusion for Object Representation"
_ICLR.cc/2025/Conference — ICLR 2025 Poster_

### Official Review · Reviewer_8Piy · 2024-11-01

**Soundness:** 2
**Presentation:** 2
**Contribution:** 2
**Rating:** 5
**Confidence:** 3

**Summary:**

This paper is motivated by Object-Centric Learning (OCL) AND introduces a multi-scale fusion scheme to vanilla VAE and improves its representation of different scales of objects. The motivation is convincing, but the technical novelty is limited and multi-scale fusion is a widely used technique in many models and computer vision tasks. Though this paper proposes some specific design modules, like multi-scale features augmentation, they are straightforward and improvements are limited according to the ablation study.
Though the technical novelty is limited, this paper investigate an interesting view, OCL, to VAE, should be benefit the comunity.

**Strengths:**

The MSF is straightforward and easy to understand. According to the experimental results, it works well and improves a strong basline model DINO.
OCL is an interesting and promising direction and this paper outperform previous OCL methods.
The method can be generalized to virous computer tasks and model to improve their representation of objects.

**Weaknesses:**

According to Sec. 4.1, the improvements from MSF are limited, just 1-2 points in many metrics.
The overall presentation is not satisfactory and difficult to understand for readers not familiar with OCL.

**Questions:**

Can the proposed method be applied to state-of-the instance segmentation models, like MaskDINO, to improve their results (~50 mask AP) on MS-COCO for instance segmentation.

---

> ### Author Response · Authors · 2024-11-22
>
> Thank you for your feedback.
>
>
>
> ### __Weakness 1__
> > According to Sec. 4.1, the improvements from MSF are limited, just 1-2 points in many metrics.
>
> We would like to clarify possible misunderstandings.
>
> Techniques OGDR and MSF in Tab. 1 are added to SLATE separately, i.e., the table contains results for SLATE+OGDR and SLATE+MSF, not SLATE+OGDR+MSF together. Therefore, MSF improves SLATE 5-7 points in both ARI and ARIfg, that is, 24.18->30.95@ARI and 24.54->30.47@ARIfg.
>
> Besides, the methods we cover are all strong baselines, and any improvement can be challenging. We exclude earlier methods like IODINE, MONet, SA and SAVi due to their lower accuracy. In OCL literature, ARI and ARIfg (2nd and 3rd columns in those tables) are widely adopted metrics, revealing more significant differences among methods.
>
>
>
> ### __Weakness 2__
> > The overall presentation is not satisfactory and difficult to understand for readers not familiar with OCL.
>
> We have improved the presentation in our paper's new version.
>
> And we would be happy to polish it further if you could provide specific examples or clarify which parts are unclear.
>
>
>
> ### __Question 1__
> > Can the proposed method be applied to state-of-the instance segmentation models, like MaskDINO, to improve their results (~50 mask AP) on MS-COCO for instance segmentation.
>
> Our MSF only modifies VAE's codebook quantization part, making it applicable to any model using codebook quantization. Unfortunately, MaskDINO does not have such a module. This requires further investigation in the future.

---

> ### Author Response · Authors · 2024-11-26
>
> Dear Reviewer 8Piy, following up on our response to your valuable feedback.
>
> Your comments have been greatly appreciated and have helped refine the work.
>
> If you had a chance to review the rebuttal, we would be grateful for any additional thoughts or clarifications, especially if there are unresolved concerns that we can address to improve the paper further.
>
> Thank you again for your time and insights, looking forward to hearing from you soon.

---

### Official Review · Reviewer_7kQN · 2024-11-02

**Soundness:** 3
**Presentation:** 3
**Contribution:** 2
**Rating:** 6
**Confidence:** 2

**Summary:**

This paper addresses the lack of multi-scale object representation in VAE-guided OCL problems by constructing a multi-scale VAE model. This enables the VAE-generated results to represent multi-scale objects, thereby improving performance across various tasks.

**Strengths:**

1. The paper is clear writing and easy to follow.

**Weaknesses:**

1. I think the paper's novelty is very limited. Although I am not familiar with object-centric learning, I believe that constructing multi-scale VAE features has already been applied in many generative tasks[1, 2]. Therefore, the core innovation of this paper, in my opinion, does not warrant a standalone publication. Can the author tell me about the difference between MSF and other methods?

2. Since I am not familiar with this field, the Area Chair (AC) and the authors may disregard my opinion if it appears to be biased.

[1]Visual Autoregressive Modeling: Scalable Image Generation via Next-Scale Prediction, arxiv2404.02905
[2]SPAE: Semantic Pyramid AutoEncoder for Multimodal Generation with Frozen LLMs, NeurIPS2023

**Questions:**

I am not familiar with this field, so please refer to the weakness

---

> ### Author Response · Authors · 2024-11-22
>
> Thank you for your feedback.
>
>
>
> ### __Weakness 1__
> > Although I am not familiar with object-centric learning, I believe that constructing multi-scale VAE features has already been applied in many generative tasks[1, 2]. Can the author tell me about the difference between MSF and other methods?
>
> Please note that we have updated the related contents into Sect. "A.1 Extended Related Work" of our paper’s new version.
>
> __Short answer__
>
> - VAR [1] vs MSF: VAR auto-regresses from smaller scales of VAE representation to the larger, but with __no information fusion among scales__. In contrast, our MSF realizes fusion among different scales on VAE for the first time.
>
> - SPAE [2] vs MSF: SPAE relies on multi-modal foundation model CLIP, while our MSF does not. SPAE element-wisely averages multiple scales into one, simply mixing different scales together, i.e., __no fusion among scales__. Our MSF augments any scaled representation with all other scales' high-quality representations, i.e., fusion among scales.
>
> __Long answer__
>
> [*__MSF__*] interpolates the input and produces multiple scales of VAE representation, which augment one another. Every scale integrates high-quality representations from the other scales, yielding better visual explanation (Fig. 3). Specifically,
>
> (1) The input image $T$ is spatially interpolated into a pyramid {$T_n$}, from high to low resolutions, and after VAE encoding, we have multiple scales of VAE intermediate representation {$Z_n$};
>
> (2) We project it into two copies, one for scale-invariant quantization using a shared codebook, yielding {$X_n^{si}$}, and the other for scale-variant quantization using different codebooks, yielding {$X_n^{sv}$};
>
> (3) We conduct our unique and novel inter-scale fusion and intra-scale fusion upon {$X_n^{si}$} and {$X_n^{sv}$}, yielding the augmented {$X_n$}, where *__whichever scale is augmented by the other scales with their high-quality representation__*, i.e., fusion among the scales;
>
> (4) The final {$X_n$} is used to guide OCL training.
>
> [*__VAR__*] interpolates the single VAE intermediate representation and produces multiple scales of VAE representations, where there is no augmentation in between. Specifically,
>
> (1) The input image $im$ is VAE encoded once, yielding one single intermediate representation $f$;
>
> (2) The authors spatially interpolate $f$ into multiple scales and residually quantize them with a shared codebook, yielding a sequence of discrete representations $R=[r_1, r_2,... r_K]$, from low to high resolutions;
>
> (3) *__There is no fusion among the scales__*;
>
> (4) Finally the authors conduct auto-regression upon $R$ for image generation, i.e., $p(r_1, r_2, ... r_k) = \Pi p(r_k | r_1, r_2,... r_{k-1})$.
>
> [*__SPAE__*] interpolates one VAE intermediate representation and yields multiple scales of VAE representations, but employs naive element-wise-average to integrate those scales into one. Specifically,
>
> (1) The input $I$ is VAE encoded once, yielding one single intermediate representation $Z$;
>
> (2) The authors conduct spatial interpolation on it and yield {$Z_l$}, and use pretrained multi-modal foundation model CLIP to residually quantize them under some prompt, producing a sequence of lexical tokens {$\hat{Z}_l$}, from low to high resolutions;
>
> (3) The authors streaming average the first $l$ scales into one representation $\hat{Z}\_l$ – *__They have no fusion among the scales; Instead, they just mix all scales together__*.
>
> (4) Finally the mixed representation is VAE decoded to produce the reconstruction or image generation.
>
> __Summary__
>
> Whether with VAE or not, it is not difficult to build multiple scales of representations; Outside of VAE, there are also many techniques to fuse multiple scales. But, __our MSF is the first to both build multiple scales upon VAE and realize fusion among multiple scales__, using unique and innovative designs based on codebook matching.
>
>
>
> ### __Reference__
>
> [1] Tian et al. Visual Autoregressive Modeling: Scalable Image Generation via Next-Scale Prediction. NeurIPS 2024.
>
> [2] Yu et al. SPAE: Semantic Pyramid AutoEncoder for Multimodal Generation with Frozen LLMs. NeurIPS 2023.

---

### Official Review · Reviewer_CqKd · 2024-11-02

**Soundness:** 3
**Presentation:** 2
**Contribution:** 2
**Rating:** 6
**Confidence:** 4

**Summary:**

This paper addresses Object-Centric Learning (OCL), which aims to capture comprehensive object information by reconstructing inputs using intermediate representations from a Variational Autoencoder (VAE). The focus is on multi-scale training, acknowledging that objects may appear at various scales in images or videos due to changes in imaging distance or intrinsic size differences. The authors propose Multi-Scale Fusion (MSF) applied to VAE representations to guide OCL for both transformer-based approaches and state-of-the-art diffusion models. The method demonstrates promising performance on datasets such as COCO, ClevrTex, and VOC.

**Strengths:**

1) The overall intuition and direction of the paper is good and deal with important problems of Object-Centric Learning.
2) The multi-scale training makes a lot sense since dealing with objects is an important problem and generally has not been solved fully in the field.
3) The paper shows good analysis and shows object separability of VAE guidance in Fig3. Fig2 quantitative results also look good.

**Weaknesses:**

1) Results on real world datasets especially COCO in Table 1 are really incremental. It’s not really clear how much advantage is by adding MSF.
2) Analysis of varying object sizes can show results on scale understanding better than overall IoU improvement.
3) Is there any intuition as to why the value of n is 3? Can we do a similar experiment on OpenImages and see if this holds true across datasets? For Openimages, if it’s easier you can try using the smaller subset of open images curated in this paper [1].
4) Discussion on the area of unsupervised semantic segmentation and object detection using SSL methods should be added. Papers like [1,2,3] should be discussed.
References:
1) Hyperbolic Contrastive Learning for Visual Representations beyond Objects CVPR2023
2) VideoCutLER
3) MOST: Multiple Object localization with Self-supervised Transformers for object discovery

**Questions:**

Overall the paper is good, but the results specially on COCO seems a bit incremental.

---

> ### Author Response · Authors · 2024-11-22
>
> Thank you for your feedback.
>
>
>
> ### __Weakness 1__
> > Results on real world datasets especially COCO in Table 1 are incremental. Not really clear how much advantage MSF adds.
>
> We would like to clarify possible misunderstandings.
>
> Techniques OGDR and MSF in Tab. 1 are added to SLATE separately, i.e., the table contains results for SLATE+OGDR and SLATE+MSF, not SLATE+OGDR+MSF together. Therefore, MSF improves SLATE 5-7 points in both ARI and ARIfg, that is, 24.18->30.95@ARI and 24.54->30.47@ARIfg.
>
> Besides, the methods we cover are all strong baselines, and any improvement can be challenging. We exclude earlier methods like IODINE, MONet, SA and SAVi due to their lower accuracy. In OCL literature, ARI and ARIfg (2nd and 3rd columns in those tables) are widely adopted metrics, revealing more significant differences among methods.
>
>
>
> ### __Weakness 2__
> > Analysis of varying object sizes can show results on scale understanding better than overall IoU improvement.
>
> We follow COCO's small/medium/large size splits to evaluate our MSF's performance, as shown in the table below. MSF shows more improvement on small and medium objects. This demonstrates that the fusion among multiple scales effectively improves different-sized object representations.
>
> Table 1. How MSF performs on different-sized objects. Dataset is COCO instance segmentation.
> |  | mIoU_S | mIoU_M | mIoU_L |
> |:---------:|:------:|:------:|:------:|
> | SLATE | 8.57 | 26.65 | 34.57 |
> | SLATE+MSF | 12.63 | 28.14 | 34.83 |
>
> Please note that the related contents have been updated into Sect. "A2. Extended Experiments" in our paper’s new version.
>
>
>
> ### __Weakness 3__
> > Is there any intuition why the value of n is 3? Can we do a similar experiment on OpenImages subset [1] and see if this holds true across datasets?
>
> We use input resolutions 128 and 256 to evaluate on OpenImages subset. Under resolution 128, we use $n$=3 and 4; under resolution 256, we use $n$=3, 4 and 5. Results are shown below. We use ARI+ARIfg as the metrics because ARI mostly reflect how well the background is segmented and ARIfg only measures foreground objects. According to the results, under resolution 128, n=3 is the best choice; under resolution 256, n=4 is the best choice. $n$ value is stable across datasets.
>
> Table 2. Effects of input resolutions and #scales $n$. Model is SLATE+MSF; dataset is OpenImages subset.
> | resolution | 128 | 128 | 256 | 256 | 256 |
> |:----------:|:-----:|:-----:|:-----:|:-----:|:-----:|
> | $n$ | 3 | 4 | 3 | 4 | 5 |
> | ARI+ARIfg | 62.07 | 60.93 | 64.82 | 67.58 | 62.75 |
>
> Please note that the related contents have been updated into "A.2 Extended Experiments" of our paper’s new version.
>
>
>
> ### __Weakness 4__
> > Discussion on the area of unsupervised semantic segmentation and object detection using SSL methods should be added. Papers like [1,2,3] should be discussed.
>
> Firstly, relationships among SSL segmentation (SSLseg), OCL, as well as World Models (WMs) are as follows. (1) __SSLseg__ focuses on extracting segmentation masks; (2) __OCL__ represents each object as a feature vector, with segmentation masks as byproducts, kind of overlapping with SSLseg; (3) __WMs__, upon OCL, address downstream tasks like visual reasoning, planning and decision-making.
>
> *__Briefly, OCL and SSLseg are designed for different purposes. OCL can be used directly to support WMs to work on visual tasks like reasoning, planning and decision-making; In contrast, SSLseg needs some intermediate step like OCL to support WMs on those advanced vision tasks.__*
>
> Please read Fig. 4 in our paper's new version, which provides a nice picture to show the relationships intuitively.
>
> Since SSLseg and OCL are designed for different purposes, __direct comparisons are not commonly discussed in the OCL literature__ – We follow the evaluation protocols established by [4, 5, 6].
>
> Please note that we have updated the related contents into our paper’s new version.
>
>
>
> ### __Reference__
>
> [1] Ge et al. Hyperbolic Contrastive Learning for Visual Representations beyond Objects. CVPR2023.
>
> [2] Wang et al. VideoCutLER: Surprisingly Simple Unsupervised Video Instance Segmentation. CVPR 2024.
>
> [3] Rambhatla. MOST: Multiple Object Localization with Self-Supervised Transformers for Object Discovery. ICCV 2023.
>
> [4] Locatello et al. Object-Centric Learning with Slot Attention. NeurIPS 2020.
>
> [5] Singh et al. Illiterate DALL-E Learns to Compose. ICLR 2022.
>
> [6] Kipf et al. Conditional Object-Centric Learning from Video. ICLR 2022.

---

### Official Review · Reviewer_a2Kd · 2024-11-03

**Soundness:** 3
**Presentation:** 3
**Contribution:** 3
**Rating:** 8
**Confidence:** 4

**Summary:**

The paper proposes multi-scale fusion (MSF) for object-centric learning. Specifically, MSF extracts VAE representations corresponding to the same input, downsampled to varying resolutions. The output representations are then upsampled and downsampled as necessary and then combined as a way to augment each set of features. The augmented features can then be fed as input to the general OCL decoder (along with the slot attention outputs and the matching entries from the discrete codebook). This intends to allow for better, more separable, semantically distinct representations of objects at varying scales. The quality is demonstrated with varying metrics corresponding to unsupervised segmentation on the masks that are generated as a byproduct of this OCL pipeline.

**Strengths:**

[S1] The method seems to consistently outperform competitors for the OCL benchmarks, and is even somewhat competitive with the foundation model baseline (DINOSAUR).

[S2] The approach is well-motivated.

[S3] Figure 3 effectively demonstrates, qualitatively, how the MSF accomplishes better OCL, that is, features that are better connected to the objects themselves.

**Weaknesses:**

[W1] The impact of the work seems limited. The ideas about multi-scale representation and fusion themselves are not novel (see for example FPN in object detection literature), but the implementation and application to this task are. However, the implementation neglects potentially more impactfully redesigns of the primary encoder-decoder pipeline or the slot attention mechanism itself.

[W2] Additionally, there are no results for any downstream tasks. Thus, while interesting, it is hard to project the impact of the work beyond this niche.

Minor: The notation and its interaction with Figure 3 is unpleasant and difficult to follow. In particular, the differences between scale-variant, scale-invariant, and representation before inter-scale fusion are quite slight and seem slightly arbitrary.

Minor: Algorithm 1 would fit in the main paper, and help add clarity in the flow.

**Questions:**

What downstream tasks could this method help with? It doesn't seem to be a SAM alternative, so what is the potential practical impact?

What would change were this applied to higher-resolution images? MSF would seem to have some promise for small objects, but those are essentially all filtered out by the pre-processing.

---

> ### Author Response · Authors · 2024-11-22
>
> Thank you for your positive feedback.
>
>
>
> ### __Weakness 1.1__
> > The ideas of multi-scale are not novel (e.g., FPN), but implementation and application to OCL are.
>
> We agree that multi-scale is a key challenge in computer vision.
>
> Existing multi-scale methods (i) Either employ image pyramids without information fusion among pyramid levels, where inter-scale information is wasted; (ii) Or rely on feature pyramids (FPN and numerous variants), where features of different layers are fused by channel-concat or element-wise-sum, mixing both low- and high-quality representations together.
>
> Although our main focus is on the OCL setting, __our MSF is the first to enable multi-scale fusion on VAE representations__. By leveraging codebook matching, we selectively fuse high-quality information among scales, rather than mixing them together. Our MSF is superior to channel-concat or element-wise-sum, as shown below.
>
> Table 1. Effects of different fusion techniques among multiple scales. Model is SLATE; dataset is ClevrTex.
> |    | channel-concat | element-wise-sum | MSF |
> |:---------:|:--------------:|:----------------:|:------:|
> | **ARI+ARIfg** | 92.63 | 89.18 | 100.71 |
>
> Please note that the related contents have been updated into Sect. "A.1 Extended Related Work" of our paper's new version.
>
>
>
> ### __Weakness 1.2__
> > The implementation neglects potentially more impactfully redesigns.
>
> We are open to including other impactful OCL designs if specific examples are provided. Regarding earlier works like IODINE, MONet, SA, and SAVi, we omitted them due to their lower performance.
>
>
>
> ### __Weakness 2__
> > No results for downstream tasks.
>
> **Downstream tasks of OCL (+MSF) include** (1) scene understanding, e.g., VisualGenome; (2) visual reasoning, e.g., Clevrer VQA; (3) visual planning, e.g., PHYRE and Physion; and (4) visual decision-making for reinforcement learning agents, e.g., video game playing and robotic manipulations.
>
> **OCL literature usually does not cover these tasks**. However, we provide results for visual planning in PHYRE with a SlotFormer [2] World Model (WM) below.
>
> Table 2. Visual planning on PHYRE [1]. OCL extracts slots; WM infers upon slots. AUCCESS [2] is the metric, the larger the better.
> | OCL | SLATE | SLATE+MSF |
> |:-------:|:----------:|:----------:|
> | **WM** | SlotFormer | SlotFormer |
> | **AUCCESS** | 84.73 | 89.95 |
>
> **Our MSF also has potential in visual generation**. This is because we only modify VAE, the key module of visual generation like StableDiffusion for images and Sora for videos. However, this is a totally different research field, beyond our scope.
>
>
>
> ### __Weakness 3__
> > Minor: The notation and its interaction with Figure 3 is difficult to follow.
> > The differences between scale-variant, scale-invariant, and representation before inter-scale fusion are quite slight.
>
> We have improved Fig. 3 presentation, especially the figure caption, in our paper’s new version. Please check the new pdf file for details.
>
> And we are happy to improve further. Could you kindly provide specific examples?
>
>
>
> ### __Weakness 4__
> > Minor: Algorithm 1 would fit in the main paper, and help add clarity in the flow.
>
> Thank you for your kind advice. We have updated our paper’s new version accordingly.
>
>
>
> ### __Question 1__
> > What downstream tasks could this method help with? It doesn't seem to be a SAM alternative, so what is the potential practical impact?
>
> Segmentation models like SAM only extract masks, but cannot represent objects as corresponding feature vectors, i.e., slots. In contrast, OCL can do both.
>
> With slots from OCL, an object-centric world model can be built for downstream tasks like visual understanding, reasoning, planning and decision-making [2]. Our MSF improves OCL, thus helps with all of these tasks.
>
>
>
> ### __Question 2__
> > What would change were this applied to higher-resolution images? MSF would seem to have some promise for small objects.
>
> We follow COCO's Small/Medium/Large size splits to evaluate our MSF's performance. Results of resolution 128x128 and 256x256 are shown below. Our MSF does show better performance on small objects; And switching to higher-resolution does not change the conclusion.
>
> Table 3. How MSF performs on different-sized objects. Dataset is COCO instance segmentation.
> | resolution 128 | mIoU_S | mIoU_M | mIoU_L |
> |:----------------:|:--------:|:--------:|:--------:|
> | SLATE | 8.57 | 26.65 | 34.57 |
> | SLATE+MSF | 12.63 | 28.14 | 34.83 |
>
> | resolution 256 | mIoU_S | mIoU_M | mIoU_L |
> |:----------------:|:--------:|:--------:|:--------:|
> | SLATE | 13.64 | 28.76 | 35.37 |
> | SLATE+MSF | 16.04 | 29.98 | 35.88 |
>
> Please note that the related contents are updated into Sect. "A.2 Extended Experiments" of our paper’s new version.
>
>
>
> ### __Reference__
>
> [1] Bakhtin et al. PHYRE: A New Benchmark for Physical Reasoning. arXiv:1908.05656.
>
> [2] Wu et al. SlotFormer: Unsupervised Visual Dynamics Simulation with Object-Centric Models. ICLR 2023.

---

> > ### Comment · Reviewer_a2Kd · 2024-11-25
> > **Considering improving rating**
> >
> > The response to weakness 1.1 seems to be a rephrasing of my original statement. The ablation in the new table is helpful, but only in a minor sense, since it doesn't affect the original criticism. Regarding 1.2, this is a fair point, and my original concern was overly speculative. Consider it withdrawn.
> >
> > For weakness 2, I appreciate the comparison to SlotFormer for planning. While I would prefer a stronger focus on such comparisons (as claims about downstream tasks are empty without the concrete evidence, very often "the devil is in the details"), this does help address my concern somewhat.
> >
> > For weakness 3, the notation for scale-invariant representations, scale-variant representations, and representations before
> > inter-scale fusion appear unchanged, and this was the root of my initial complaint. To be clear, I'm talking about the notation, not the visual content of the figure.
> >
> > For question 1, see weakness 2.
> >
> > For question 2, the answer and expanded results are helpful.

---

### Author Response · Authors · 2024-11-26

We thank all the reviewers for their insightful feedback and responding to our rebuttal.

We have now colored all changes with a blue color in the paper, where in the main content the updates mainly save space and improve the presentation while the appendix contains now extended related work and experiments suggested by the reviewers.

For those reviewers who did not yet have a chance to respond to our rebuttal, please let us know before the paper revision deadline if you would like us to do further improvements to the paper.

---

### Meta-Review · Area_Chair_LPhk · 2024-12-20

**Metareview:**

This paper explores Object-Centric Learning (OCL), which seeks to capture comprehensive object information by leveraging intermediate representations from a Variational Autoencoder (VAE) to reconstruct inputs. The approach emphasizes multi-scale training, recognizing that objects in images or videos may appear at different scales due to variations in imaging distance or intrinsic size disparities.

The average score of this paper is 6.25 (8, 6, 6, 5), which is above the borderline. After carefully checking the response during rebuttal period and reading the paper, I decide to give the acceptance recommendation.

Note that the authors are required to update/revise/polish their paper according to reviewers' suggestions in the final version.

**Additional Comments On Reviewer Discussion:**

The average score of this paper is 6.25 (8, 6, 6, 5), which is above the borderline. Reviewer 8Piy points out that the obtained gains are small (1~2 points), however, without any further valuable reviews. This review should be ignored according to the policy of ICLR 2025.

After carefully checking the response during rebuttal period and reading the paper, I decide to give the acceptance recommendation.

---

### Decision · Program_Chairs · 2025-01-22

Accept (Poster)